


# Ozone affected by a succession of four landfall typhoons in
# the Yangtze River Delta, China: major processes and health
# impacts
Chenchao Zhan [a,1], Min Xie [a,*], Chongwu Huang [a,1], Tijian Wang [a], Jane Liu [b,c], Meng Xu [d], Chaoqun
Ma [a], Jianwei Yu [e], Yumeng Jiao [f], Mengmeng Li [a], Shu Li [a], Bingliang Zhuang [a], Ming Zhao [a],
Dongyang Nie [a]
[a] School of Atmospheric Sciences, Joint Center for Atmospheric Radar Research of CMA/NJU,
CMA-NJU Joint Laboratory for Climate Prediction Studies, Jiangsu Collaborative Innovation
Center for Climate Change, Nanjing University, Nanjing 210023, China
[b] College of Geographic Sciences, Fujian Normal University, 350007, Fuzhou, China
[c] Department of Geography and Planning, University of Toronto, Toronto, Ontario, Canada
[d] Jiangsu Provincial Climate Center, Nanjing 210009, China
[e] Jiangsu Provincial Meteorological Observatory, Nanjing 210008, China
[f] Department of Microbiology and Parasitology, Bengbu Medical College, Bengbu 233030, China

15 --------------------------------------------------------------------

[*] Corresponding author. minxie@nju.edu.cn (M. Xie)
[1] The third author can be considered as the co-first author
**Abstract:** Landfall typhoon can significantly affect $O_3$ in the Yangtze River Delta (YRD) region. In
this study, we investigate a unique case characterized by two multiday regional $O_3$ pollution
episodes related to four successive landfall typhoons in the summer of 2018 in the YRD. The results
show that $O_3$ pollution episodes mainly occurred during the period from the end of typhoon and the
arrival of the next typhoon. The moment that typhoon reached the 24-h warning line and the last
moment of typhoon activity in the mainland China can be roughly regarded as time nodes.
Meanwhile, the variations of $O_3$ was related to the track, duration and landing intensity of the
typhoons. The impact of typhoon on $O_3$ was like a wave superimposed on the background of high
$O_3$ concentration in the YRD in summer. When typhoon was near the 24-h warning line before it
landed the coast line of the YRD, the prevailing wind originally from the ocean changed to from the
inland, and transported lots of precursors from the polluted areas to the YRD. With typhoon, the low
temperature, strong upward airflows, more precipitation and wild wind prevented high $O_3$ episodes.





After typhoon, the air below the 700 hPa atmospheric layer was warm and dry, which was
conductive to the formation of $O_3$ from the abundance of precursors. It is note-worthy that $O_3$ is
mainly generated in the middle of boundary layer (~ 1000 m), and then transported to the surface
by downward airflows or turbulences. Moreover, $O_3$ can be accumulated and trapped on the ground
due to the poor diffusion conditions because the vertical diffusion and horizontal diffusion were
suppressed by downward airflows and light wind, respectively. The premature mortalities attributed
to $O_3$ exposure in the YRD during the study period is 194.0, more than the casualties caused directly
by the typhoons. This work enhances our understanding of how landfall typhoons affect $O_3$ in the
YRD, which can be helpful to forecast the $O_3$ pollution synthetically impacted by the subtropical
high and typhoon.
**Key Words:** ozone; landfall typhoon; the Yangtze River Delta region;

**1 Introduction**
The tropospheric ozone ($O_3$), which is formed by a series of complex photochemical reactions
between volatile organic compounds (VOCs) and nitrogen oxides ($NO_x=NO+NO_2$) in combination
with sunlight (Chameides and Walker, 1973; Xie et al., 2014), has received continuous attention due
to its negative impact on air quality (Chan and Yao, 2008; Monks et al., 2015), human health (Jerrett
et al., 2009), climate (Allen et al., 2012; IPCC, 2014) and biosphere (Dingenen et al., 2009).
Research on urban $O_3$ pollution can date back to the early 1950s, beginning with the Los Angeles
smog. In China, the photochemical smog, which is characterized by high level of $O_3$, was first
discovered in Xigu district of Lanzhou in 1970s (Tang et al., 1989). However, with the key
atmospheric environmental problem was coal-smoke pollution (such as acid rain) at that time (Wang
et al., 2019), little systematic research and coordinated $O_3$ monitoring were performed in China until
the mid-2000s (Wang et al., 2017).
Since the beginning of 21st century, the complex air pollution, which is dominated by fine
particulate matter ($PM_{2.5}$, particles of 2.5 microns or less in aerodynamic diameter) and surface $O_3$,
has been ingrained in the megacities of China (Chan and Yao, 2008; Jin et al., 2016; Kan et al.,
2012). Air pollution has evolved into a political and economic concern in China. Due to drastic air
pollution control since 2013, particle pollution has been greatly reduced, appearing a significantly
decrease in sulfur dioxide ($SO_2$), $NO_x$ and $PM_{2.5}$. However, the concentrations of $O_3$ and VOCs had



an increase from 2013 to 2017 ( Li et al., 2017), suggesting that more attention should be paid to
controlling $O_3$ and VOCs in the future. Overall, the causes of air pollution and the control policies
are still a major challenge in China, especially in understanding the sources, transport and dispersion
processes, and chemical formation mechanisms of $O_3$ and its precursors (Ding et al., 2016; Guo et
al., 2014; Huang et al., 2014).
Typhoon (tropical cyclone, TC) is one of the most important factors of natural disasters in East
Asia. Out of the total provinces in China, 10 coastal and 6 island provinces are affected by typhoon
induced disasters, with more than 250 million lives are affected (Liu et al., 2009). The average
number of typhoons making landfall in China is 9 each year, and those typhoons usually inflict vast
losses in human life and property due to the accompanied strong wind, torrential rains and huge
storm surges (Zhang et al., 2009; Zhao et al., 2012). Because of the long lifetime and tremendous
energy, typhoon has a significantly influence on local atmospheric conditions, and thereby can affect
surface $O_3$ concentration through advection, diffusion, deposition and other processes. The impact
of typhoon on $O_3$ has attracted extensive attention in recent years (Deng et al., 2019; Huang et al.,
2005; Shu et al., 2016; Wang and Kwok, 2002; Wei et al., 2016; Yang et al., 2012). For example,
Deng et al. (2019) reported that high $O_3$ and high aerosol concentrations (double high episodes) are
likely to occur when the PRD is under the control of typhoon periphery and subtropical high with
strong downdrafts. Previous studies were mainly in the southern China (including Hong Kong and
Taiwan), where are frequently affected by typhoons. However, research on the impact of landfall
typhoons on $O_3$ is still limited.
The Yangtze River Delta (YRD) region, being one of the most developed and densely
populated regions in China, is located on the western coast of the Pacific Ocean. With 3.7% of the
area and 16.0% of the population of China, the YRD contributed over 20% of the national total
Gross Domestic Product (GDP) in 2019. Due to the rapid economic development and high energy
consumption, this region is suffering from intense air pollution (Ding et al., 2013; Li et al., 2019;
Wang et al., 2015; Xie et al., 2016). In 2017, the 90th percentile of the maximum daily 8-hour
average (MDA8) $O_3$ concentration was 170 μg m$^{-3}$, and 16 of the 26 cities (Figure 1b) in the YRD
failed to meet national standard (http://www.cnemc.cn/jcbg/zghjzkgb/201905/t20190529_704755.
html). Therefore, it is an urgent task to investigate the spatiotemporal characteristic of $O_3$ as well as
its formation mechanism in the YRD. Influenced by monsoon weather, the warm and stagnation



conditions play an important role in the occurrence of high-level $O_3$ in summer (Li et al., 2018; Liao
et al., 2015; Lu et al., 2018; Zhao et al., 2010). Synoptic weather systems, such as typhoon and cold
front, also have significant effects on $O_3$ in the YRD (Hu et al., 2013; Shu et al., 2016). This work
aims to reveal the main processes of landfall typhoon affecting surface $O_3$ in the YRD, hoping to
fill the knowledge gap and be helpful for making reasonable pollution control measures.

In this study, we report an outstanding case observed in the YRD during the period from 16

July to 25 August, during which multiday episode of high $O_3$ occurred and was related to four
successive landfall typhoons. Base on the monitoring data and numerical simulation, we further
explore the impact of landfall typhoons on $O_3$ in the YRD, including the major processes and health
impacts. The following part of this paper is structured as the follows: Section 2 gives a brief
description of monitoring data, the analysis methods, and model configurations. The results as well
as the discussions are detailed in section 3. Section 4 summarizes the main conclusions.

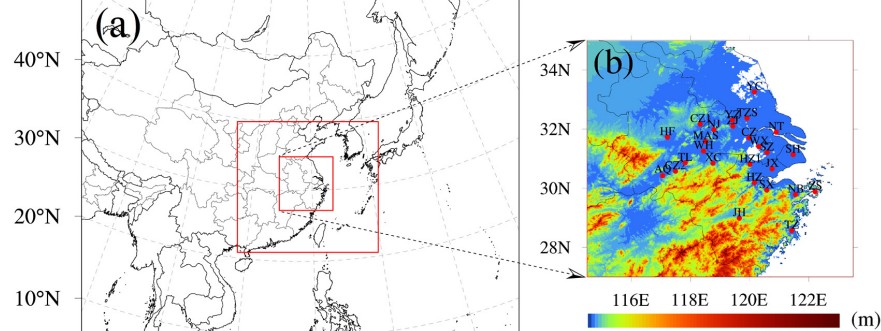


**Figure 1. The three nested modeling domains (a) in WRF, and the locations of 26 typical cities**
**in the YRD with terrain elevation data (b). The cities in (b) include: Nanjing (NJ), Wuxi (WX),**
**Changzhou (CZ), Suzhou (SZ), Nantong (NT), Yancheng (YC), Yangzhou (YZ), Zhenjiang (ZJ)**
**and Taizhoushi (TZS) located in Jiangsu province; Hangzhou (HZ), Ningbo (NB), Jiaxing (JX),**
**Huzhou (HZ1), Shaoxing (SX), Jinhua (JH), Zhoushan (ZS) and Taizhou (TZ) located in**
**Zhejiang province; Hefei (HF), Wuhu (WH), Maanshan (MAS), Tongling (TL), Anqing (AQ),**
**Chuzhou (CZ1), Chizhou (CZ2) and Xuancheng (XC) located in Anhui province; and the**





**megacity Shanghai (SH). The terrain elevation data are available at**
**https://www.ngdc.noaa.gov/mgg/global/relief/ETOPO1/data/bedrock/cell_registered/netcdf/.**

**2 Data and methods**
**2.1 Air quality data**
Surface air pollutants monitored by the China National Environmental Monitoring Center
(CNMC) Network are used in this study. The nationwide observation network began operating in
74 major cities in 2013, and it included 1597 nonrural sites covering 454 cities by 2017 (Lu et al.,
2018). The monitoring data are strictly in accordance with the national monitoring regulations
(http://www.cnemc.cn/jcgf/dqhj/), and can be acquired from the national urban air quality real-time
publishing platform (http://106.37.208.233:20035/). Each monitoring site automatically measures
hourly air pollutants ($PM_{2.5}$, $PM_{10}$, $SO_2$, $NO_2$, $O_3$ and CO), and the urban hourly pollutants are
calculated by averaging the pollutants measured at all monitoring sites in that city. The MDA8 $O_3$
is calculated based on the hourly $O_3$ with more than 18-h measurements (Liao et al., 2017). Manual
inspection, including the identification and handling of invalid and lacking data, is performed
following previous studies (Xie et al., 2016; Shu et al., 2017; Zhan et al., 2019).
**2.2 Surface and sounding meteorological data**
With respect to surface observed meteorological data, stations at the three provincial capital
cities (Nanjing, Hangzhou and Hefei) and the megacity Shanghai are selected, which are ZSNJ
(32.00°N, 118.80°E), ZSHC (30.23°N, 120.17°E), ZSOF (31.87°N, 117.23°E) and ZSPD (31.12°N,
121.77°E), respectively. These surface observations, including 2-m temperature, 10-m wind speed
and direction and 2-m relative humidity, are recorded hourly and can be obtained from the website
of the University of Wyoming (http://weather.uwyo.edu/surface/).
To verify the upper-air fields, the sounding observations at Shanghai (31.40°N, 121.46°E) and
Nanjing (32.00°N, 118.80°E) are used. These sounding observations (pressure, temperature, relative
humidity, wind direction and wind speed etc.) are also acquired from the website of the University
of Wyoming (http://weather.uwyo.edu/upperair/sounding.html), with a time resolution of 12 h
(00:00 and 12:00 UTC).
**2.3 The best-track TC dataset**
To capture the characteristics of landfall typhoons, the best-track TC dataset issued by the



China Meteorological Center (CMA) is considered due to its good performance on the landfall
typhoons in the mainland China (available at http://tcdata.typhoon.org.cn/zjljsjj_sm.html). The
dataset covers seasons from 1949 to the present, the region north of the equator and west of 180°E,
and is updated annually (Li and Hong, 2016; Ying et al., 2014). A wealth of information on typhoon
is recorded every 6h in the dataset, including location, minimum sea level pressure, etc. For landfall
typhoons, 24h before they land and during their activities in the mainland China, the data will be
recorded every 3h. Refer to the national standard for grade of tropical cyclones (GB/T 19201-2006),
the intensity category (IC) of tropical cyclones is given in the dataset, which is based on the near
surface maximum 2-min mean wind speed near the tropical cyclone center, ranges from 1 to 6 (Table

1).


**Table 1. The intensity category of tropical cyclones**

| Intensity category (IC) | The near surface maximum 2-min mean wind speed near the tropical cyclone center (m/s) | Beaufort scale |
|---|---|---|
| Tropical depression (IC=1) | 10.8-17.1 | 6-7 |
| Tropical storm (IC=2) | 17.2-24.4 | 8-9 |
| Severe tropical storm (IC=3) | 24.5-32.6 | 10-11 |
| Typhoon (IC=4) | 32.7-41.4 | 12-13 |
| Severe typhoon (IC=5) | 41.5-50.9 | 14-15 |
| Super typhoon (IC=6) | $\geq 51.0$ | $\geq 16$ |


**2.4 Model description and configurations**
To simulate the high $O_3$ episodes over the YRD during the period with typhoons, the WRF-
CMAQ one-way coupled model is applied, which consists of WRF v3.6.1
(https://www2.mmm.ucar.edu/wrf/users/) developed by the United States National Center for
Atmospheric Research (NCAR) and CMAQ v5.0.2 (https://github.com/USEPA/CMAQ/tree/5.0.2)
developed by the United States Environmental Protection Agency (EPA).
WRF generates offline meteorological inputs for CMAQ with initial and boundary conditions
from the National Centers for Environmental Prediction (NCEP) global final analysis fields every





6 h at a spatial resolution of 1° × 1° (https://rda.ucar.edu/datasets/ds083.2/). Three nested domains
are used, with horizontal resolutions of 81, 27 and 9 km, and grids of 88 × 75, 85 × 79 and 97 × 97,
respectively (Figure 1a). There are 24 vertical sigma layers from surface to 100 hPa, with about 8
layers located below 1.5 km to resolve the boundary layer processes. Furthermore, the major
physical options for the dynamic parameterization in WRF are summarized in Table 2.

**Table 2. The domains and physical options for WRF in this study**

| Items | Contents |
|---|---|
| Dimensions (x, y) | (88, 75), (85, 79), (97, 97) |
| Grid spacing (km) | 81, 27, 9 |
| Microphysics | WRF Single-Moment 5-class scheme (Hong et al., 2004) |
| Longwave radiation | RRTM scheme (Mlawer et al., 1997) |
| Shortwave radiation | Goddard scheme (Kim and Wang, 2011) |
| Surface layer | Moni-Obukhov scheme (Monin and Obukhov, 1954) |
| Land-surface layer | Noah land-surface model (Chen and Dudhia, 2001) |
| Planetary boundary layer | YSU scheme (Hong et al., 2006) |
| Cumulus parameterization | Grell-Devenyi ensemble scheme (Grell and Devenyi, 2002) |


Since the horizontal domains of CMAQ are one grid smaller than WRF, all three nested
domains are adjusted automatically. The vertical layers of CMAQ are the same as WRF. The
Meteorology Chemistry Interface Processor (MCIP) can convert WRF outputs to the necessary
meteorological inputs for CMAQ. Moreover, the CB05 gas-phase mechanism with aqueous/cloud
chemistry is selected in the CMAQ configurations.
The anthropogenic emissions are from the Multi-resolution Emission Inventory for China
(MEIC) in 2016 with the resolution of 0.25° (http://meicmodel.org/), including anthropogenic
emissions from power generation, industry, agriculture, residential and transportation sectors. All
emission estimates are spatially allocated to the relevant grid cells based on the meteorological fields
obtained from WRF, and are temporally distributed on an hourly basis. The simulation starts from
00:00 UTC on 13 July to 00:00 UTC 27 August, with the first 72 h as spin-up time.





**2.5 Integrated process rate (IPR) analysis**

To quantify the contributions of individual processes in $O_3$ formation, the IPR analysis provided in the CMAQ is utilized. The IPR analysis can illustrate the contributions to changes in pollutant concentrations from seven different types of processes, including horizontal advection (HADV), vertical advection (ZADV), horizontal diffusion (HDIF), vertical diffusion (VDIF), dry deposition (DDEP), cloud processes with the aqueous chemistry (CLDS) and chemical reaction process (CHEM), with a mass conservation adjustment at each model grid cell. The IPR analysis has been widely applied to investigate regional air pollution (Fan et al., 2015; Li et al., 2012; Wang et al., 2010). In this study, MADV is defined as the sum of HADV and ZADV, and TDIF is defined as the sum of HDIF and VDIF.

**2.6 Model evaluation**

To evaluate the model performance, the simulation results in the innermost domain, including $O_3$ concentration, air temperature at 2 m ($T_2$), relative humidity (RH), wind speed at 10 m ($WS_{10}$) and wind direction at 10 m ($WD_{10}$), are examined against the hourly observations at the representative cities (Table 3). The statistical metrics, including correlation coefficient (R), root-mean-square error (RMSE) and normalized mean bias (NMB), are used in this study. They are defined as follows:

$$R = \frac{\sum_{i=1}^{N}(S_i - \overline{S})(O_i - \overline{O})}{\sqrt{\sum_{i=1}^{N}(S_i - \overline{S})^2}\sqrt{\sum_{i=1}^{N}(O_i - \overline{O})^2}} \ , \tag{3}$$

$$RMSE = \sqrt{\frac{\sum_{i=1}^{N}(S_i - O_i)^2}{N}} \ , \tag{4}$$

$$NMB = \frac{\sum_{i=1}^{N}(S_i - O_i)}{\sum_{i=1}^{N}O_i} \times 100\% \ , \tag{5}$$

where $S_i$ and $O_i$ are the simulations and observations, respectively. N is the total number of valid data. $\overline{S}$ and $\overline{O}$ are the average value of simulations and observations. In general, the model results are acceptable if the values of R, RMSE and NMB are close to 1, 0 and 0, respectively (Li et al.,





2017; Shu et al., 2016; Xie et al., 2016).

**2.7 Estimate of health impacts**

Previous studies showed that surface $O_3$ pollution can induce a series of adverse health impacts
by causing the incidence and mortality of respiratory diseases (Ghude et al., 2016; Jerrett et al.,
2009; Lelieveld et al., 2015). To arouse more attention on the issue that $O_3$ can be significantly
affected by typhoons in the YRD, we further estimate the premature mortality attributed to $O_3$ during
the study period.
A standard damage function (Anenverg et al., 2010; Liu et al., 2018; Voorhees et al., 2014;
WS/T 666-2019, Technical specifications for health risk assessment of ambient air pollution of
China) is employed to quantify premature mortality due to $O_3$ exposure:
$$\Delta M = y_0 (\frac{RR-1}{RR}) Pop \ ,  \tag{1}$$
where $\Delta M$ is the excess mortalities attribute to $O_3$ exposure, $y_0$ is the baseline mortality rate, RR is
relative risk and (RR-1)/RR is the attributable fraction, and Pop is the exposed population. RR can
be calculated using the following relationship:
$$RR = \exp(\beta(C - C_0)) \ ,  \tag{2}$$
where $\beta$ is the concentration-response factor, C is the exposure concentration and $C_0$ represents the
theoretical minimum-risk concentration.
In this study, the mortality rate for respiratory disease is obtained from China Health and
Family Planning Statistical Yearbook 2018 (https://www.yearbookchina.com/navibooklist-
n3018112802-1.html), which is 68.02/100000. The $\beta$ is generated from Dong et al. (2016), that is
0.461%. The population data are obtained from the bureau of statistics of different cities in the YRD.
The $C_0$ is 70 µg m$^{-3}$ for MDA8 $O_3$ given by the World Health Organization (WHO).

**3 Results and discussions**

**3.1 Characteristic of $O_3$ episodes**

In the midsummer season, the warm sea surface (high temperature) is conducive to the
generation of typhoon (high $O_3$ concentration), which provides a good opportunity to investigate
the mechanism of typhoon affecting $O_3$ in the YRD. Figure 2 shows the MDA8 $O_3$ in the typical 26
cities of the YRD in summer of 2018. $O_3$ concentration tended to be high in June, and relatively low




in July and August. The relatively low $O_3$ may be attributed to the maritime air masses transported
by the Asian summer monsoon (Ding et al., 2008; Xu et al., 2008). Nevertheless, we notice that
there are two regional multiday $O_3$ pollution episodes from 24 July to 11 August in the YRD, which
means that about half of the cities in the YRD exceed the national air quality standard (The national
ambient air quality standard for MDA8 $O_3$ is 160 μg m$^{-3}$ in China). The first multiday $O_3$ episodes
appeared in most cities from 24 July to 2 August. The highest MDA8 $O_3$ concentration reached up
to 264 μg m$^{-3}$ on 27 July in Ningbo (NB). $O_3$ pollution was even observed for 6 consecutive days
from 27 July to 1 August in Maanshan (MAS). Only two days later, regional $O_3$ pollution occurred
in the YRD again from 5 August to 11 August.

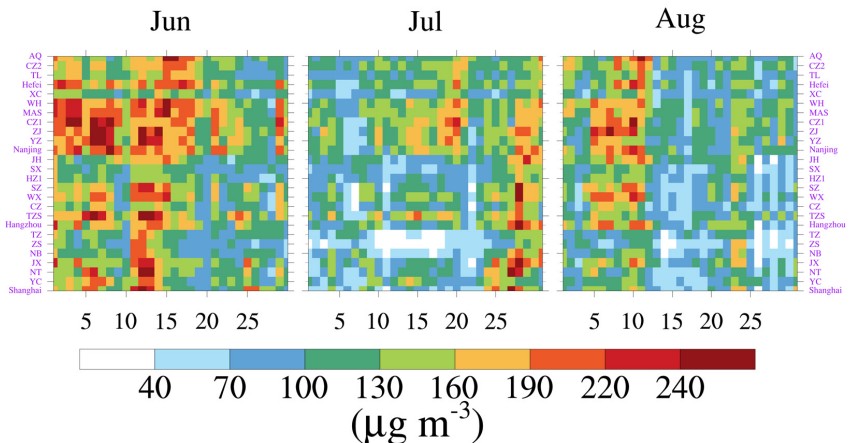


**Figure 2. The MDA8 $O_3$ in 26 cities of the YRD in summer of 2018. The national ambient air**
**quality standard for MDA8 $O_3$ is 160 μg m$^{-3}$ in China. These cities are sorted by longitude.**

Figure 3 further shows diurnal variation of $O_3$ in all 26 cities of the YRD from 00:00 16 July

to 00:00 25 August (throughout this paper the time refers to UTC, unless LST is specifically
mentioned). Interestingly, $O_3$ pollution occurred earlier in cities near the coastline (large longitude,
Figure 1b) rather than concurrently during the two multiday $O_3$ episodes. For example, from 24 July
to 2 August, the first day that hourly $O_3$ concentration exceeded the national air quality standard
(The national ambient air quality standard for hourly $O_3$ is 200 μg m$^{-3}$ in China) in Shanghai,
Hangzhou, Nanjing and Hefei was 24 July, 27 July, 28 July and 31 July, respectively. Thus, we



classify the 26 cities in the YRD into four categories based on their longitudes, surrounding the four
representative cities (Figure 4). The category I cities include SH, YC, NT, JX, NB, ZS and TZ. The
category II cities include HZ, TZS, CZ, WX, SZ, HZ1, SX and JH, and the category III cities include
NJ, YZ, ZJ, CZ1, MAS, WH and XC. Other cities are classified as the category IV cities, which are
HF, TL, CZ2 and AQ. The first category cities are closest to the coastline, while the fourth category
is the opposite.

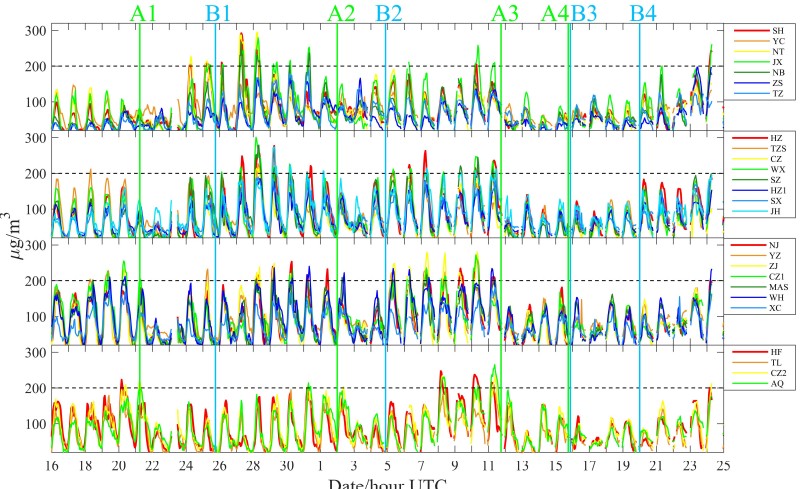


**Figure 3. Diurnal variation of O₃ in all 26 cities of the YRD. The grey dotted lines are the**
**national ambient air quality standard for hourly O₃ (200 μg m⁻³) in China. The letter A**
**indicates the moment that the typhoon has reached the 24-h warning line, and letter B**
**indicates the last moment of typhoon activity in the mainland China. These moments are**
**acquired from the best-track TC dataset, depending on the start and end time of the densified**
**observations. Coordinates 1, 2, 3, and 4 represent Typhoon Ampil, Typhoon Jongdari,**
**Typhoon Yagi, and Typhoon Rumbia, respectively. Note: these cities are sorted by longitude.**

**3.2 Landfall typhoons and their effects**
For such O₃ episodes with regional, long-lasting characteristics, may often be associated with
slow-moving synoptic weather systems. We find that the O₃ episodes coincided well with landfall
typhoons activities, and the track and intensity of typhoons are given in Figure 4. Typhoon Ampil





was first observed at 00:00 on 18 July, and landed in Shanghai around 4:30 on 22 July with an
intensity of severe tropical storm (IC=3). During the time of Typhoon Ampil, Typhoon Jongdari
generated over the western North Pacific at 12:00 on 23 July, and made landfall at the junction of
Zhejiang province and Shanghai at 21:00 1 August. After Typhoon Jongdari, Typhoon Yagi
generated at 00:00 7 August. At around 15:35 12 August, it landed in Zhejiang province and
remained active in the mainland China until 21:00 15 August. Before the end of Typhoon Yagi,
Typhoon Rumbia was observed over the western North Pacific at 6:00 14 August. It finally landed
in Shanghai at around 20:00 16 August, causing huge economic losses.

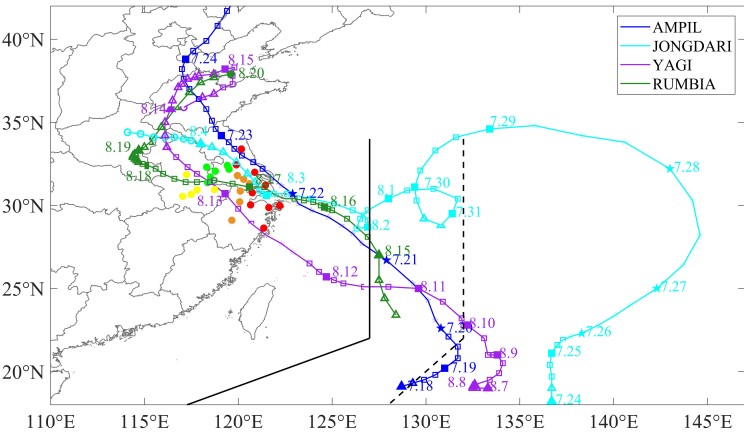


**Figure 4. The track and intensity of Typhoon Ampil, Typhoon Jongdari, Typhoon Yagi, and**
**Typhoon Rumbia. The circle, triangle, square and pentagram indicate the intensity category**
**of tropical cyclones is less than 1 (IC < 1), equal to 1 (IC = 1), equal to 2 (IC = 2), and not less**
**than 3 (IC >= 3), respectively. Black solid line and dotted line represent the 24-hour and 48-h**
**warning line for tropical cyclones, respectively. The colored solid points are the locations of**
**cities in the YRD, and different color represents different cities categories. Wherein, red,**
**carrot, green and yellow are category I, II, III and IV cities, respectively.**

To further understand the relationship between O$_3$ episodes and landfall typhoons, we mark the
critical moments of landfall typhoons in Figure 3. The letter A indicates the moment that typhoons
have reached the 24-h warning line, and the letter B indicates the last moment of typhoon activity



in the mainland China. These moments are acquired from the best-track TC dataset, depending on
the start and the end time of the densified observations. Coordinates 1, 2, 3, and 4 represent Typhoon
Ampil, Typhoon Jongdari, Typhoon Yagi, and Typhoon Rumbia, respectively. As shown in Figure
3, $O_3$ exhibited a significant cycle during the study period. That is, when the typhoon is close enough
(near moments A1, A2, A3 and A4), the $O_3$ concentrations decreased, but $O_3$ concentrations would
increase as long as the typhoon was not active in the mainland China (B1, B2 and B4) any more.
This cycle would repeat if the next typhoon approached. $O_3$ pollution was likely to occur during the
period from the end of the typhoon to the arrival of the next typhoon (B1A2 and B2A3) in the YRD.
Furthermore, we find that the variations of $O_3$ was related to the track, duration and landing
intensity of the typhoons. For example, during the B1A2 period when the $O_3$ pollution occurred, the
moments that hourly $O_3$ concentrations first exceed 200 μg m$^{-3}$ in about half of cities of the
categories I, II, III and IV were 6:00 UTC (14:00 LST) 27 July, 6:00 UTC (14:00 LST) 28 July, 3:00
UTC (11:00 LST) 29 July and 6:00 UTC (14:00 LST) 31 July, respectively. This phenomenon also
suggests that $O_3$ pollution occurs in coastal region will be ahead of that in inland regions, which
may be related to the track of typhoons (Figure 4). As for the impact of typhoon duration, the A4B3
period provided a good interpretation. While Typhoon Yagi was still active in the mainland China,
Typhoon Rumbia had reached the 24-hour warning line. Hence, the $O_3$ remained a low level
throughout the period (A3B4), which was quite different from B1A2 and B2A3 period. Noted that
the landing point and active path of Typhoon Ampil and Typhoon Jongdari were very similar (Figure
4). However, the landing intensity of Typhoon Ampil was severe tropical storm (IC = 3), and that of
Typhoon Jongdari was tropical storm (IC = 2), resulting in a difference in $O_3$ concentrations for
Shanghai. Within 24 hours after Typhoon Ampil (Jongdari) reached the 24-hour warning line, the
average $O_3$ concentrations was 40.9 (80.1) μg m$^{-3}$ in Shanghai. This is because that the stronger the
typhoon landed, the gale (The 10-m wind speed near moment A1 was larger than that near moment
A2 in Shanghai, Figure 7a) and precipitation accompanying the typhoon will be more effective in
removing $O_3$.
**3.3 Processes of $O_3$ pollution affecting by typhoons**
To reveal the major processes of $O_3$ pollution episodes affected by landfall typhoons, four
representative cities (Shanghai, Hangzhou, Nanjing and Hefei) are selected for further analysis –
based on monitoring data and model results.



### 3.3.1 Evaluation of model performance

To evaluate the simulation performance, the hourly simulation results are compared with the measurements during 00:00 16 July to 00:00 25 August. Table 3 presents the statistical metrics for selected variables, including temperature at 2 m ($T_2$), relative humidity (RH), wind speed at 10 m ($WS_{10}$) and wind direction at 10 m ($WD_{10}$), and surface $O_3$. $T_2$ is reasonably well simulated, with R values of 0.75, 0.77, 0.72 and 0.64 in Shanghai, Hangzhou, Nanjing and Hefei, respectively. Though our simulation underestimates $T_2$ to some certain extent, the slightly underestimation is acceptable due to the small RMSE (3.2, 2.7, 2.9 and 3.3) and NMB (-7.5%, -5.1%, -5.5% and -5.5%) values. As for RH, the simulation results are overestimated in all four cities, leading to the NMB values of 9.1%, 4.6%, 6.7% and 0.5% in Shanghai, Hangzhou, Nanjing and Hefei, respectively. With high R values (0.69, 0.65, 0.71 and 0.71) and relatively low RMSE values (12.4, 12.8, 12.1 and 10.8), the WRF simulates RH over the YRD quite well. The wind fields are closely related to the transport processes of air pollutants. The overestimation of $WS_{10}$ may partly be attributed to the unresolved terrain features by the default surface drag parameterization causing overestimation of wind speed in particular at low values (Jimenez and Dudhia, 2012; Li et al., 2017). With regards to $WD_{10}$, the simulation error is large based only on these statistical metrics. This is because that near-surface wind fields are deeply influenced by local underlying surface characteristics, and improving the urban canopy parameters might be useful (Liao et al., 2015; Xie et al., 2016). In term of $O_3$, the simulation results for $O_3$ concentrations behave satisfactorily. R is as high as 0.55, 0.65, 0.66 and 0.54 in Shanghai, Hangzhou, Nanjing and Hefei, respectively, while the NMB values are 5.8%, 16.4%, -6.2% and -5.3%, respectively.

**Table 3. Statistical metrics for meteorological and chemical variables.**

| City | Variable | $\bar{O}$ | $\bar{S}$ | R | RMSE | NMB |
|------|----------|------|------|------|------|------|
| Shanghai | $T_2$ (℃) | 30.3 | 28.1 | 0.75 | 3.2 | -7.5% |
| | RH (%) | 75.0 | 81.8 | 0.69 | 12.4 | 9.1% |
| | $WS_{10}$ (m s$^{-1}$) | 4.9 | 5.5 | 0.51 | 2.3 | 11.7% |
| | $WD_{10}$ (°) | 144.8 | 113.4 | 0.01 | 113.5 | -22.9% |
| | $O_3$ (μg m$^{-3}$) | 74.3 | 76.5 | 0.55 | 45.3 | 5.8% |
| Hangzhou | $T_2$ (℃) | 30.3 | 28.8 | 0.77 | 2.7 | -5.1% |
| | RH (%) | 75.1 | 78.5 | 0.65 | 12.8 | 4.6% |
| | $WS_{10}$ (m s$^{-1}$) | 3.3 | 4.7 | 0.32 | 2.7 | 32.5% |





| | | | | | | |
|---|---|---|---|---|---|---|
| | WD$_{10}$ (°) | 155.0 | 114.7 | -0.10 | 132.5 | -27.8% |
| | O$_3$ (µg m$^{-3}$) | 81.7 | 91.3 | 0.65 | 49.8 | 16.4% |
| Nanjing | T$_2$ (℃) | 29.8 | 28.1 | 0.72 | 2.9 | -5.5% |
| | RH (%) | 77.4 | 82.6 | 0.71 | 12.1 | 6.7% |
| | WS$_{10}$ (m s$^{-1}$) | 3.1 | 5.0 | 0.39 | 3.0 | 63.8% |
| | WD$_{10}$ (°) | 132.8 | 115.6 | 0.21 | 102.7 | -15.0% |
| | O$_3$ (µg m$^{-3}$) | 87.6 | 79.8 | 0.66 | 46.7 | -6.2% |
| Hefei | T$_2$ (℃) | 29.3 | 27.7 | 0.64 | 3.3 | -5.5% |
| | RH (%) | 81.1 | 81.5 | 0.71 | 10.8 | 0.5% |
| | WS$_{10}$ (m s$^{-1}$) | 3.2 | 3.2 | 0.37 | 2.2 | 2.9% |
| | WD$_{10}$ (°) | 147.0 | 128.6 | 0.04 | 136.7 | -13.3% |
| | O$_3$ (µg m$^{-3}$) | 87.3 | 80.3 | 0.54 | 45.0 | -5.3% |

*Note.* R exceeds 0.1 to reach statistically significant at 99.9% confident level.

Figure 5 further shows hourly variations of O$_3$, T$_2$, WS$_{10}$ and WD$_{10}$ for measurements and

simulations in four representative cities. The simulations effectively reproduce the diurnal variation
of O$_3$, T$_2$ and WS$_{10}$, confirming the reliability of the simulation results. Moreover, the model well
captures the shift in wind direction during the study period. Thus, the overall model performance
for wind fields is acceptable. In summary, the simulation results can capture and reproduce the major
characteristics of the O$_3$ episodes, including the meteorological conditions and evolution of O$_3$,
which can provide valuable insights into the formation of the O$_3$ episodes.

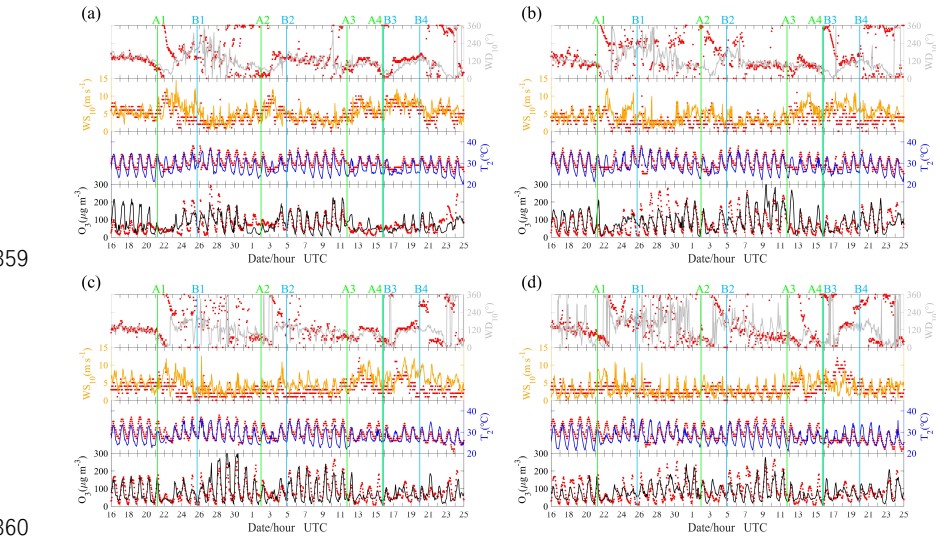





**Figure 5. Hourly variations of $O_3$, $T_2$, $WS_{10}$ and $WD_{10}$ for measurements (red dots) and**
**simulation (colored lines) in (a) Shanghai, (b) Hangzhou, (c) Nanjing and (d) Hefei.**

**3.3.2 Shanghai in category I cities**
In this study, Shanghai was usually one of the first cities affected by landfall typhoons. We can
see a multiday episode of $O_3$ during the period of 24-28 July, with a maximum of hourly $O_3$ up to
292 µg m$^{-3}$ at 27 July (Figure 6a). The high $O_3$ concentrations together with high primary pollutants
(CO and $NO_2$) suggest a strong photochemical $O_3$ production under the condition of high
temperature (The daily maximum temperature can reach 35 ℃) during this period, and the weak
wind may play a significant role in the accumulation of surface $O_3$. The increase of primary
pollutants may be related to the wind shift from southeast to southwest causing by Typhoon Ampil
(A1 in Figure 6a, -A1 and A1B1 in Figure 7), resulting in air masses originally from the ocean had
become inland. Interestingly, $PM_{2.5}$ also showed good correlation with $O_3$ and primary pollutants,
especially for $NO_2$ during this period. This indicates that a high level of oxidizability can promote
the formation of secondary particles (Kamens et al., 1999; Khoder, 2002). From the results of
process analysis (Figure 6b), the major contributions to surface $O_3$ were TDIF, CHEM and DDEP
due to the small net contribution of MADV. TDIF had a considerable positive contribution while
DDEP did the opposite, suggesting that high surface $O_3$ may be sourced from the upper layer via
TDIF process, and be removed via DDEP process. However, for the whole boundary layer, which
is defined as the layer less than 1500 m in this study, it was the balance between CHEM and DDEP
instead TDIF and DDEP. Thus, TDIF was likely to play the role of "transport" from the upper layer
to surface. Figure 6c further shows the temporal-vertical distribution of $O_3$ with vertical wind
velocity. The downward airflows were prevailed over Shanghai until 23 July, which are induced by
the subtropical high. Then, strong upward airflows appeared as Typhoon Ampil arrived, and high
level of $O_3$ disappeared. Around 27 July, the downward airflows gradually resumed and high level
of $O_3$ occurred. It is note-worthy that the high value center of $O_3$ appeared near the altitude of 1 km
instead of near surface, indicting high photochemical production efficiency of $O_3$ occurred in the
middle boundary layer. The downward airflows can not only inhibit the vertical transport of $O_3$ but
also transport high-level $O_3$ to the surface, causing the episodes of surface $O_3$.
As shown in Figure 7, $O_3$ pollution tends to occur during the period from the end of the typhoon





to the arrival of the next typhoon (B1A2 and B2A3) in the YRD. To reveal this phenomenon, we
compare these two periods (B1A2 and B2A3) with their previous periods (A1B1 and A2B2) using
the skew-T log-P diagram (Figure 6d and 6e). It is found that the atmospheric conditions of B1A2
(B2A3) were hotter and drier than A1B1 (A2B2) below 700 hPa in Shanghai, and wind speed is
smaller in B1A2 (B2A3). Those changes in atmospheric conditions after typhoon will be conducive
to the generation of high $O_3$ concentration in Shanghai.

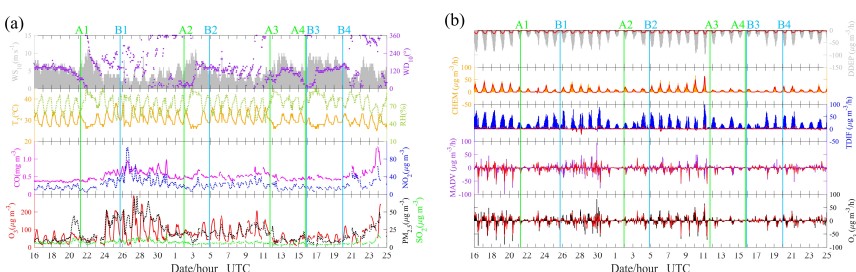


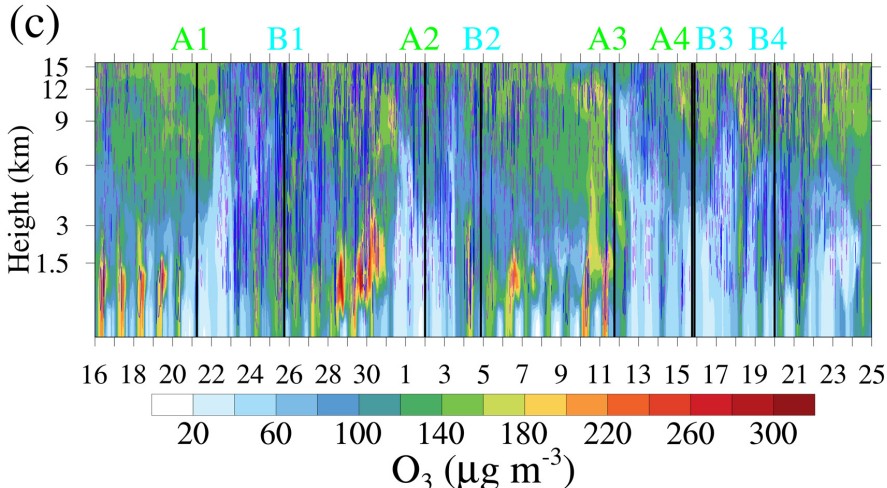




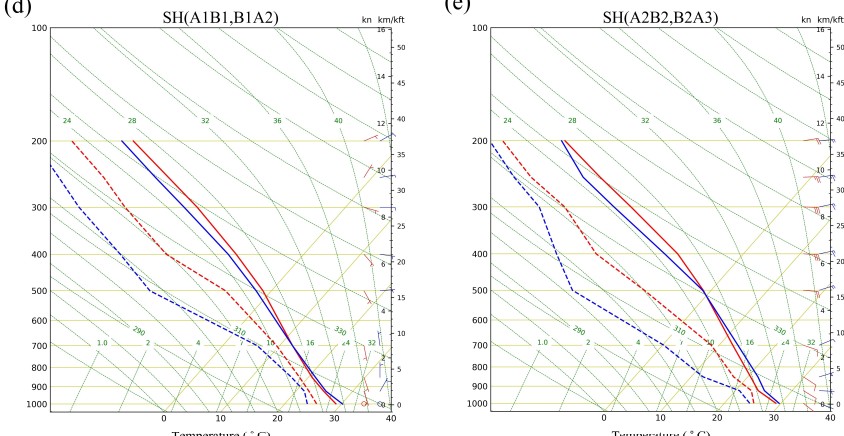


**Figure 6. (a) Time series of air pollutants ($O_3$, $PM_{2.5}$, $SO_2$, CO and $NO_2$) and meteorological**

**factors ($T_2$, RH, $WS_{10}$ and $WD_{10}$) in Shanghai. (b) Individual processes contribution to net $O_3$**

**density at Shanghai. $O_3$ is the net increase, MADV is the sum of horizontal advection (HADV)**

**and vertical advection (ZADV), TDIF is the sum of horizontal diffusion (HDIF) and vertical**

**diffusion (VDIF), CHEM is chemical reaction process, and DDEP is dry deposition process.**

**The color histograms indicate the results for the layer near the surface, while the solid red**

**lines indicate the average results for all layers below 1500 m. (c) Temporal-vertical distribution**

**of $O_3$ with vertical wind velocity over Shanghai. The dotted purple line and solid blue line**

**indicate the negative wind speeds (downward airflows) and positive wind speeds (upward**

**airflows), respectively. (d) The skew-T log-P diagram at Shanghai. The average results of**

**period A1B1 and B1A2 are shown in red and blue, respectively. (e) Same as (d), but for the**

**average results of period A2B2 and B2A3.**

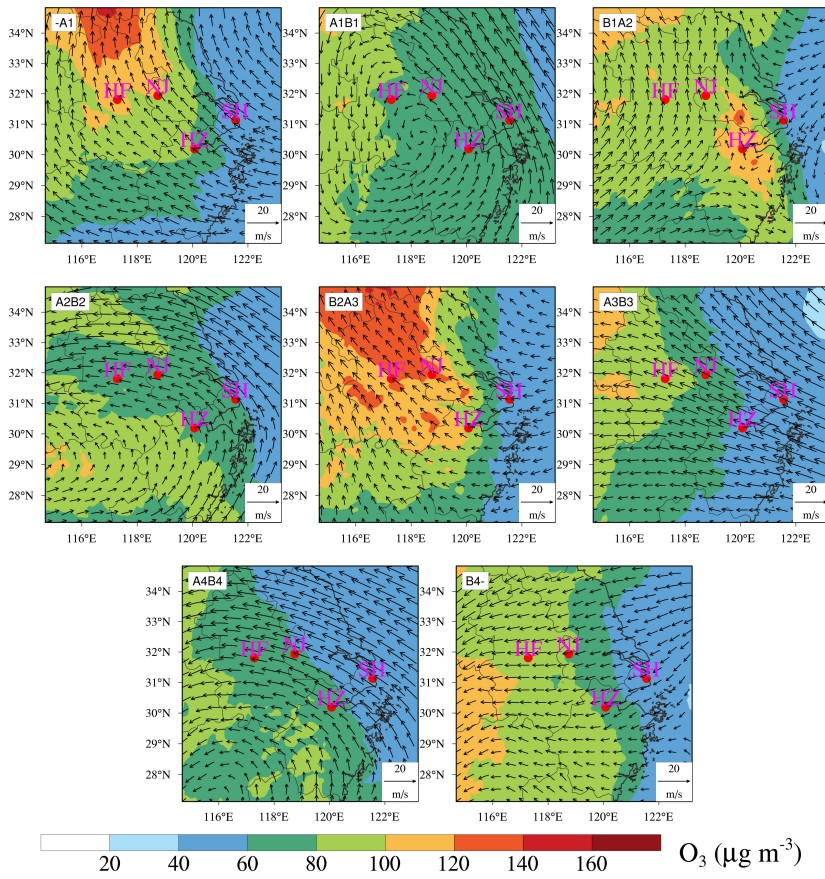

**Figure 7. Spatial distribution of surface $O_3$ with wind fields at 850 hPa over the YRD. -A1, A1B1, B1A2, A2B2, B2A3, A3B3, A4B4, B4- are the average results from the beginning to A1, A1 to B1, B1 to A2, A2 to B2, B2 to A3, A3 to B3, A4 to B4, B4 to the end, respectively. Details can be found in Figure 4.**

### 3.3.3 Hangzhou in category II cities

Figure 8 presents the results for Hangzhou. It shows that high $O_3$ concentrations occurred on 27-31 July and 5-7 August, which may also be related to the strong photochemical production of $O_3$ under the abundance of precursors (Figure 8a) and poor diffusion conditions due to the light wind (B1A2 and B2A3 in Figure 7). Figure 8a further shows that high $O_3$ was often associated with an increase in CO but the $NO_2$ concentrations usually remained at the same level. This phenomenon indicates a VOCs-limited regime in this city since CO usually have good correlation with VOCs



and can play a similar role as VOCs in the photochemical production of $O_3$ (Atkinson, 2000; Ding
et al., 2013). In fact, $O_3$ in other representative cities (Shanghai, Nanjing and Hefei) also showed a
better correlation with CO than $NO_2$. Though Hangzhou is close to Shanghai, there is a significant
difference of wind fields over these two cities. Starting from the arrival of Typhoon Ampil (A1).
The wind direction in Hangzhou did not change back to southeast until a few days later after
Typhoon Jongdari dissipated (B2). During this period (A1B2), the frequent southwest wind may be
the reason for high CO concentrations in Hangzhou. In addition, the chaotic wind field during period
B1A2 (B1A2 in Figure 7) may lead to the light wind in Hangzhou. With respect to the simulation
results, the model simulated the variation of $O_3$ but failed to capture the $O_3$ peaks (e.g., the peak
values on 27-31 July), which may be related to the strong upward airflows (Figure 8c) that inhibited
the accumulation of $O_3$ (Figure 8b). This further illustrates that downward airflows may be an
important factor for $O_3$ episodes in this case.


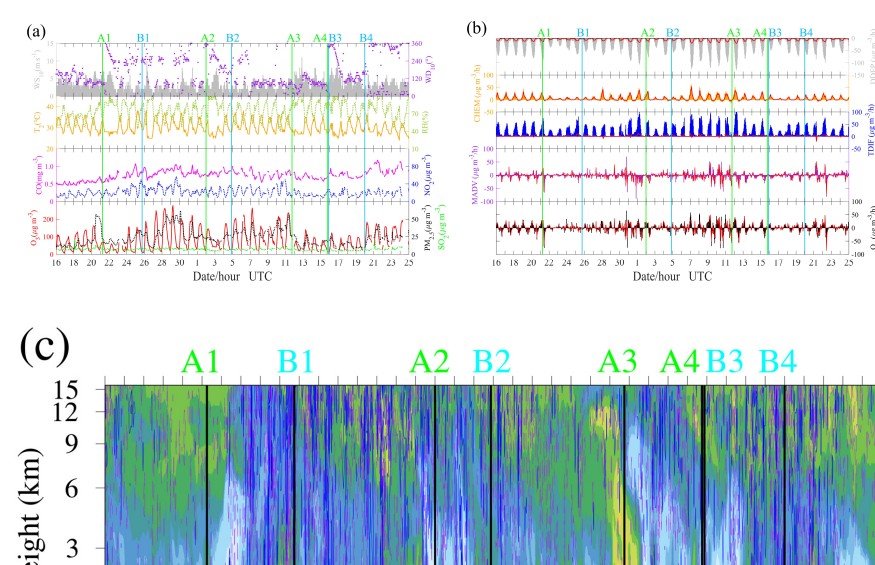






**Figure 8. Same as Figure 6 (a)-(c), but for Hangzhou.**

### 3.3.4 Nanjing in category III cities

As for Nanjing, the O$_3$ episodes with exceedance of national air quality standards were observed on 28 July to 1 August and 7-11 August. These O$_3$ episodes were characterized by abundant O$_3$ precursors under the condition of high temperature. Furthermore, light wind (B1A2 and B2A3 in Figure 7) and downward airflows (Figure 9c) also contributed greatly to the occurrence of O$_3$ pollution, with the similar mechanism as that of Shanghai and Hangzhou. As early as 22 July, the wind direction in Nanjing had changed from southeast to southwest affected by Typhoon Ampil, and the concentrations of the main primary pollutants (CO, NO$_2$ and SO$_2$) increased (Figure 9a). However, high-level O$_3$ episodes did not occur until 28 July even though the maximum temperature did not change significantly during 24-31 July. The "obstacle" of the O$_3$ episodes may be the precipitation causing by the strong upward airflows during 23-26 July (Figure 9c). As shown in Figure 9b, high surface O$_3$ concentration during the pollution episodes is the result of TDIF and CHEM processes, and is lost through DDEP and MADV processes. With respect to the vertical structure of atmospheric, B1A2 (B2A3) was also hotter and drier than A1B1 (A2B2) below 700 hPa in Nanjing (Figure 9d and 9e). The similar results as Shanghai further confirm that high O$_3$ concentrations are more likely to occur during the period from the end of the typhoon to the arrival of the next typhoon (B1A2 and B2A3) than the period when the typhoon approaches and is active in the mainland China (A1B1 and A2B2).

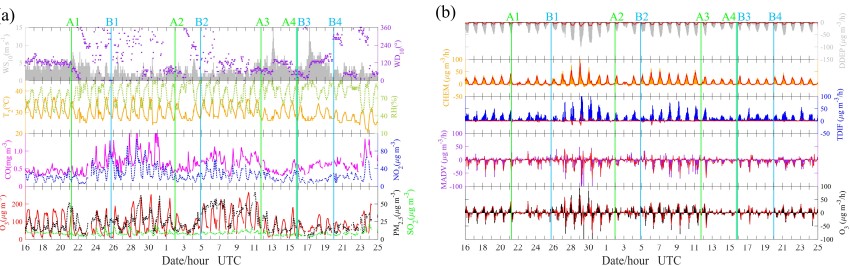

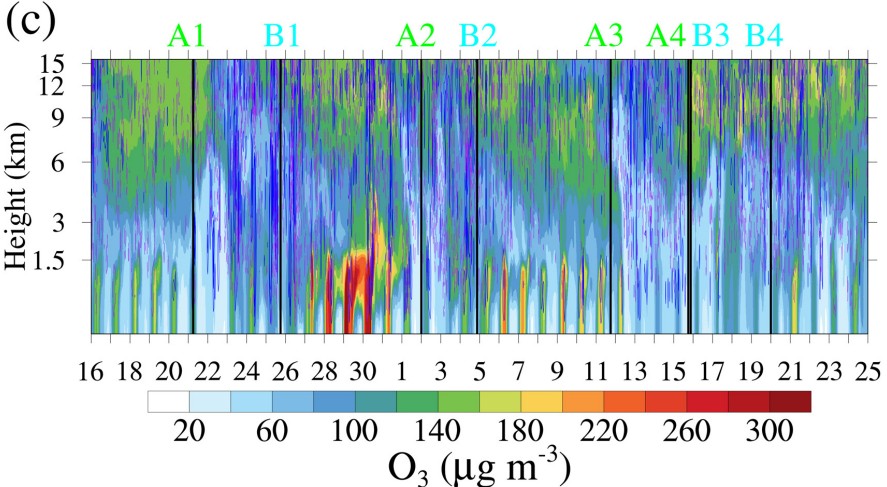

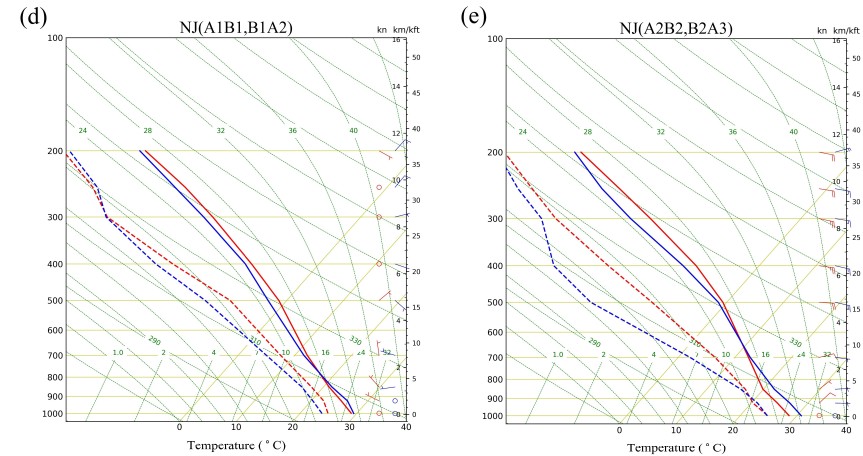

**Figure 9. Same as Figure 6, but for Nanjing.**

### 3.3.5 Hefei in category IV cities

Hefei is the city farthest from the coast among four representative cities, and O$_3$ pollution occurred on 31 July and 8-11 August. We can also find the phenomenon that the precursors concentrations had an increase once the wind direction changed from southeast to southwest (Figure 10a). During B1A2 and B2A3, the main precursors of O$_3$ had a high level. However, high O$_3$ concentration was mainly found in B2A3, and not in B1A2. This may be related to the relatively low temperature during B1A2 (Figure 10a), which is not conducive to photochemical production of



O$_3$ (Figure 10b). As shown in Figure 10c, there were distinct upward airflows within the boundary
layer, which may be related to urban effect (e.g., urban heat islands). These upward airflows within
the boundary layer help mix the air, resulting in a uniform distribution of O$_3$ in the vertical direction.
However, the downward airflows can still inhibit the vertical diffusion of O$_3$, and O$_3$ tends to be
trapped within the boundary layer.

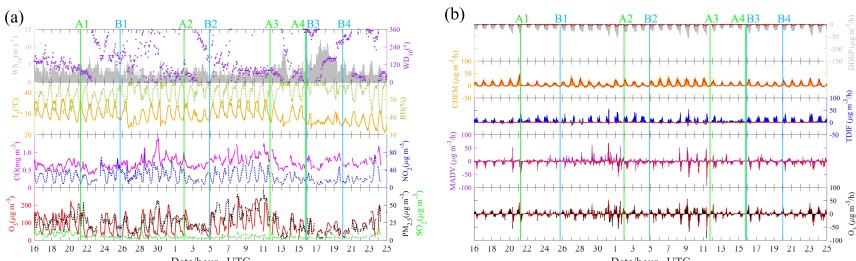


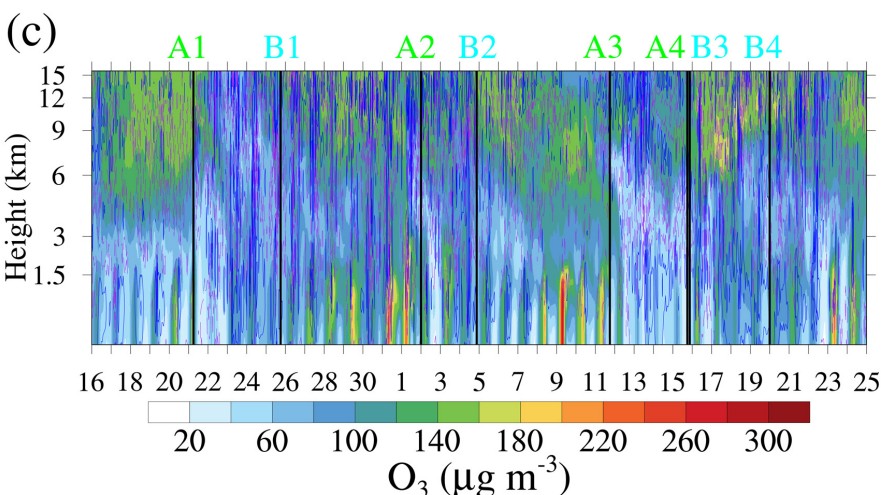


**Figure 10. Same as Figure 6 (a)-(c), but for Hefei.**

**3.3.6 A schematic diagram of major processes**

Although the processes of landfall typhoon affecting O$_3$ varied from city to city, the major

processes have many similarities and can be summarized as a schematic diagram in Figure 11. The
YRD region, which features a typical subtropical monsoon climate, is strongly influenced by the
western Pacific subtropical high in summer. Dominated by the subtropical high, the meteorological





conditions of high temperature and downward airflows combined with high levels of precursors due
to the huge energy consumption tend to form high $O_3$ concentrations in this region. However,
powerful systems like typhoon can break this state. For typhoons that may land in the YRD, by the
time they approach the 24-hour warning line, the prevailing southeast wind in the YRD will change
to southwest wind, which can transport lots of precursors from inland to the YRD. The change in
wind direction depends on the track of typhoon and the geographical location of cities, and often
appears first in coastal region. With typhoon, the low temperature, precipitation (upward airflows)
and wild wind will prevent high $O_3$ and $PM_{2.5}$ episodes. Moreover, the effect of removing pollutants
is related to the intensity of typhoon landing, but some of the main precursors of $O_3$ are still at a
high level due to foreign sources superposed with local emissions. After typhoon, the atmosphere
returns to a warm and dry state (even more so than before), and strong photochemical reactions
begin to produce $O_3$ under the abundance of precursors. $O_3$ is mainly generated inside the boundary
layer (~1000 m), and then transported to the surface by downward airflows or turbulent mixing. At
the same time, the wind readjusts to southeast and wind speed is light, resulting in poor diffusion
conditions. The downward airflows and light wind obstruct the vertical and horizontal diffusion of
$O_3$, leaving $O_3$ trapped on the ground. The thermal-dynamic effects result in high-level surface $O_3$
in the YRD.

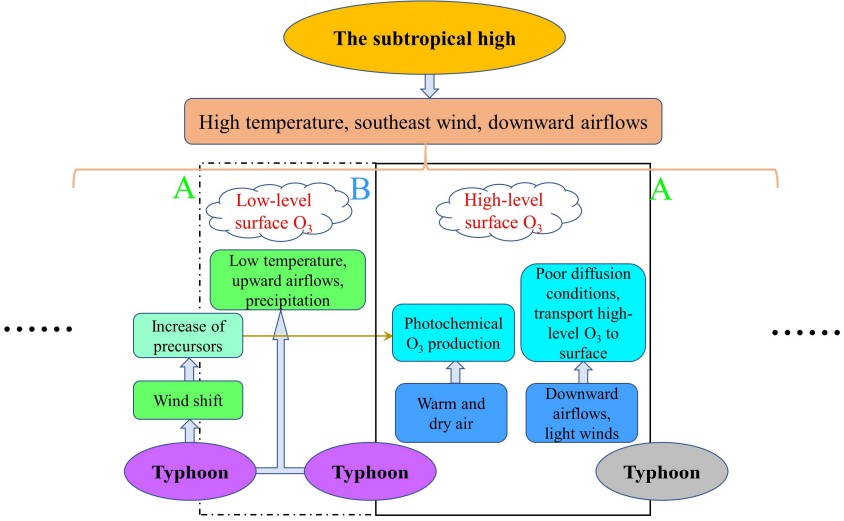




**Figure 11. A schematic diagram of major processes that summertime O₃ affected by landfall**
**typhoon in the YRD. The letter A indicates the moment that the typhoon has reached the 24-**
**h warning line, and letter B indicates the last moment of typhoon activity in the mainland**
**China.**

In fact, most typhoons generated over the western North Pacific will not land in China, or they
are more likely to land in the South China rather than the YRD. In our previous study (Shu et al.,
2016), the typhoon did not land in the YRD, but the processes of high-level $O_3$ formation may be
common. That is, the processes in the dashed box in Figure 11 are unique to landfall typhoons, while
the processes in the solid box can be found as long as the typhoons that can affect the YRD.
Transport of precursors, downward airflows, high temperature and light wind are crucial factors,
and the roles of those factors play in $O_3$ episodes depends on typhoon and city. It is hard to quantify
these processes with just a few cases. For example, we cannot estimate whether the downward
airflows are dominated by the subtropical high or the periphery circulation of typhoons since they
usually occur simultaneously. Furthermore, the behave of particulate matter is intriguing since high-
level $PM_{2.5}$ often occurs with high-level $O_3$ after typhoon, which is different from previous studies
that high particulate matter concentrations inhibit the formation of $O_3$ (Li et al., 2005; Xing et al.,
2017). This may be related to the heterogeneous reactions (Lou et al., 2014) but research on this
issue is quite limited to date.

**3.3 Premature mortalities induced by O₃ exposure**
When it comes to typhoons, especially landfall typhoons, the first concern is the huge damage
caused by extreme weathers. After typhoons, people are relieved and busy with their work as usual.
However, our research indicates that high $O_3$ episodes are likely to occur in the short period after a
typhoon in the YRD, and high $O_3$ concentrations can do harm to people's health. To arouse attention
on this issue, we estimate the premature mortality attributed to $O_3$ for respiratory disease, we choose
two complete cycles, which is the period A1A3 (21 July to 11 August), to do the calculation. Table
4 summarized the premature mortalities in cities in the YRD. The premature mortalities are a
function of both the population and $O_3$ levels, resulting in high premature mortalities in populated
and heavily polluted areas. Out of the 26 cities in the YRD, Shanghai shows highest premature





mortalities (29.2) due to its high surface $O_3$ concentrations and huge population. The city with the
lowest premature mortalities (0.6) is Zhoushan, which may be related to removing effect of the
maritime air masses as Zhoushan is located by the sea (Figure 1b). During this period, the total
premature mortalities in the YRD is 194.0, which is larger than the number of casualties caused
directly by the typhoons (80 people were killed by landfall typhoons in mainland China in 2018).

**Table 4. Premature mortalities induced by $O_3$ exposure for respiratory disease**

| | City | Population (thousand) | Premature mortalities |
|---|---|---|---|
| Category I cities | Shanghai | 24,240 | 29.2 |
| | Yancheng | 7,200 | 6.1 |
| | Nantong | 7,310 | 7.9 |
| | Jiaxing | 4,726 | 7.3 |
| | Ningbo | 8,202 | 8.1 |
| | Zhoushan | 1,173 | 0.6 |
| | Taizhou | 6,139 | 4.1 |
| Category II cities | Hangzhou | 9,806 | 16.5 |
| | Taizhoushi | 4,636 | 5.2 |
| | Changzhou | 4,729 | 4.4 |
| | Wuxi | 6,575 | 10.7 |
| | Suzhou | 10,722 | 15.3 |
| | Huzhou | 3,027 | 2.8 |
| | Shaoxing | 5,035 | 4.7 |
| | Jinhua | 5,604 | 8.2 |
| Category III cities | Nanjing | 8,436 | 13.4 |
| | Yangzhou | 4,531 | 5.5 |
| | Zhenjiang | 3,196 | 5.3 |
| | Chuzhou | 4,114 | 5.8 |
| | Maanshan | 2,337 | 3.6 |
| | Wuhu | 3,748 | 6.2 |
| | Xuancheng | 2,648 | 2.0 |
| Category IV cities | Hefei | 8,087 | 10.9 |
| | Tongling | 1,629 | 1.7 |
| | Chizhou | 1,475 | 2.1 |
| | Anqing | 4,691 | 6.4 |
| Total | | 154,016 | 194.0 |


**4 Conclusions**



548   In this study, we investigate the detail processes of landfall typhoons affecting $O_3$ in the YRD

549 based on a unique case during 16 July to 25 August with the help of monitoring observations and

550 numerical simulations. This case was characterized by two multiday reginal $O_3$ pollution episodes

551 concerned with four successive landfall typhoons. The two $O_3$ episodes appeared from 24 July to 2

552 August and 5 to 11 August, respectively, with the highest MDA8 $O_3$ reached up 264 $\mu g\ m^{-3}$.

553   The moment that typhoon reaches the 24-h warning line and the last moment of typhoon

554 activity in the mainland China are crucial, because $O_3$ pollution episodes mainly occurred during

555 the period from the end of typhoon and the arrival of the next typhoon in the YRD. These two

556 moments can be roughly regarded as time nodes. Furthermore, it is found that the variations of $O_3$

557 was related to the track, duration and landing intensity of the typhoons during the study period. $O_3$

558 pollution appeared in coastal region was ahead of that in inland regions due to the track of typhoons.

559 The interval between two typhoons can affect the duration of high $O_3$ concentration in the YRD.

560 Generally, sustained high $O_3$ concentration tends to appear on days when the typhoon has dissipated

561 but not influenced by the new one. As for the landing intensity of typhoon, the stronger the typhoon

562 landed, the gale and precipitation accompanying the typhoon will be more effective in removing $O_3$,

563 resulting in lower $O_3$ concentration in the typhoon landing location.

564   The detail processes of landfall typhoons affecting $O_3$ depend on typhoons and cities. High

565 temperature and downward airflows dominated by the subtropical high combined with abundant

566 precursors are the main reason for high $O_3$ concentration in the YRD in summer. And landfall

567 typhoon can change this state through the following mechanism: When the landfall typhoon is close

568 enough (~ 24-hour warning line), the prevailing southeast wind will change to southwest wind,

569 which transports large amount of precursors from inland to the YRD. The southwest wind usually

570 appears first in coastal region, and will turn back to southeast wind as long as the YRD is dominated

571 by subtropical high. Then the typhoon makes landfall, the low temperature, precipitation (upward

572 airflows) and wild wind suppress the generation of $O_3$. After typhoon, the atmosphere at low layer

573 (below 700 hPa) will be warm and dry, and strong photochemical reactions begin to produce $O_3$

574 under the abundance of precursors due to foreign sources superposed with local emissions. $O_3$ is

575 mainly generated in the middle of boundary layer (~ 1000 m), and then transported to the surface

576 by downward airflows or turbulent mixing. The downward airflows also obstruct the vertical

577 diffusion of $O_3$. Meanwhile, wind speed is light when the wind readjusts to southeast, which further



worsens horizontal diffusion of $O_3$. The $O_3$ can be accumulated and trapped on the ground. The
thermal-dynamic effects results in high surface $O_3$ concentration in the YRD. Those processes will
repeat if the next typhoon approach.
The estimated premature mortalities attributed to $O_3$ exposure for respiratory disease in the
YRD during 21 July to 11 August (two complete cycles of typhoons) is 194.0, which is larger than
the number of casualties caused directly by the typhoons. This work can enhance our understanding
of how landfall typhoons affect $O_3$ in the YRD, which may help to forecast the $O_3$ pollution
synthetically impacted by the subtropical high and typhoon. Meanwhile, our results further confirm
that large-scale synoptic weather systems play an important role in regional air pollution, suggesting
a need in establishing potential links between air pollution and predominant synoptic patterns.

*Author contributions.* C. C. Zhan and M. Xie had the original ideas, designed the research, collected
the data, and prepared the original draft. C. C. Zhan carried out the data analysis. M. Xie acquired
financial support for the project leading to this publication. C. W. Huang taught and helped C. C.
Zhan to do the numerical simulation. T. J. Wang and J. Liu revised the manuscript and helped to
collect the data. C. Q. Ma helped to deal with the emission inventory. M. Xu and J. W. Yu helped to
collect the data. M. M. Li, S. Li, B. L. Zhuang, and M. Zhao reviewed the initial draft and checked
the English of the original manuscript. Y. M. Jiao and D. Y. Nie reviewed the initial draft and helped
to improve the work of health impact.

*Acknowledgements.*
This work was supported by the National Key Research and Development Program of China
(2018YFC0213502, 2018YFC1506404), the open research fund of Chongqing Meteorological
Bureau (KFJJ-201607) and the Fundamental Research Funds for the Central Universities

(020714380047).

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
