# Peer review of "Ozone affected by a succession of four landfall typhoons in"

_Atmospheric Chemistry and Physics, 2020_

## Referee Comment (RC1) · Anonymous Referee #1 · 14 Sep 2020

General comments This research is talking about an interesting topic. Ozone pollution episodes caused by the landfall typhoon were analyzed based on typical cases in the Yangtze River delta in China. This study provided an insight into the characteristics of the occurrence of ozone pollution and the changes in O3 concentrations during the special synoptic system. I think this paper is well-organized and presenting an important schematic diagram.

Specific commentsïijŽ Here are some questions listed in the below, which need further addressing in the modified version:

1ãĂĄ Section 3.1, Figure2, since the influences of the typhoon cases are special in

2018, could the authors add summertime ozone concentrations in 2017 and 2019 (if the measurement data could be available) to compare with that in 2018 to show the roles clearer? 2ãĂĄ Section 3.2, Line 309, Here the authors mentioned the "coastal" region and "inland" region, what is the distance definition for them? Could you describe them here? 3ãĂĄ Section 3.3, Line 324, the representatives of the chosen cities should be addressed here, is it according to the distance off coastal lines? 4ãĂĄ Section 3.3 Table 3, what is the situation for simulated precipitation compare with the observed one? Also, t 5ãĂĄ Line 386-388, the authors mentioned the high value center of O3 appeared near altitude of 1km instead of near surface caused by the high photochemical production efficiency of ozone. What is the physical transport role in the high O3 here? 6ãĂĄ Section 3.7, the evaluation on the premature mortalities induced by O3 exposure are important, but here the authors did not give detailed methodologies or any reference about the estimation of premature mortalities. More details should be added here. Technical correctionsïjŽ 1ãĂĄ Table 3: the authors need indicating o and s with "observation" and "simulation" as a note.

---

## Referee Comment (RC2) · Anonymous Referee #2 · 17 Sep 2020

**Comments**

A typhoon over the Western Pacific, as a large scale weather system during East Asian summer monsoon season, can significantly affect the tropospheric O3 as well as air quality in a large region. The manuscript investigated the Landfall typhoon significantly affecting O3 in the Yangtze River Delta (YRD) region, East China in respects of major processes and health impacts. This unique case study presented the interesting results on major processes of $O_3$ change driven by a succession of four landfall typhoons. This topic fits well into the scope of ACP. The manuscript should be considered for publication only after making minor revisions as follows:

1) Please clarify the implication and limitation of this study for air quality change. A three-dimensional circulation of typhoon consists of the rotational air flow in the horizontal direction and the in-up-out-down overturning flow in the vertical direction, along which air mass near the surface can rise into thunderstorm clouds, outflowing at high levels in the UTLS and subsiding in the periphery. To better discuss the link with three-dimensional circulation of typhoon, please reference this ACP paper: Jiang, Y. Zhao, T., Liu, J., et al. Why does surface ozone peak before a typhoon landing in southeast China? . Atmos. Chem. Phys., 2015, 15, 13331-13338.

2) All Figs. (a) and (b) in Figs 6, 8 and 9 are too tiny to identify. Please improve the Figure presentations with better quality.

3) Lines 485-487:   this statement that "The YRD region, which features a typical subtropical monsoon climate, is strongly influenced by the western Pacific subtropical high in summer."   is incorrect or incomplete. The YRD, as a typical region of East Asian monsoon climate, is strongly influenced by typhoon activities over the Western Pacific.

4) In synoptics, the western Pacific subtropical high and tropical cyclone (Typhoon) over western Pacific are two large scale weather systems in

opposition to each other. Therefore, please remove "The subtropical high", and "High temperature, southeast wind, downward airflow" from Fig. 11, also please correct the corresponding discussions.

---

## Author Comment (AC1) · 30 Sep 2020

The comment was uploaded in the form of a supplement:
https://acp.copernicus.org/preprints/acp-2020-554/acp-2020-554-AC1-supplement.pdf

---

## Author Comment (AC2) · 30 Sep 2020

**A point to point response to the reviewers' comments**

Thank you for the reviewers' comments on our manuscript entitled "Ozone affected by a succession of four landfall typhoons in the Yangtze River Delta, China: major processes and health impacts" (acp-2020-554). Those constructive comments are all valuable for revising and improving our manuscript, as well as the important guiding significance to our researches. We have studies those comments carefully and have made correction which we hope to meet with approval. Here are point to point responses (in blue colored). Accordingly, we also revised manuscript (in red colored).

**Anonymous Referee #2:**

**General comments**

A typhoon over the Western Pacific, as a large scale weather system during East Asian summer monsoon season, can significantly affect the tropospheric $O_3$ as well as air quality in a large region. The manuscript investigated the Landfall typhoon significantly affecting $O_3$ in the Yangtze River Delta (YRD) region, East China in respects of major processes and health impacts. This unique case study presented the interesting results on major processes of $O_3$ change driven by a succession of four landfall typhoons. This topic fits well into the scope of ACP.

Response: We would like to express our great appreciation to you for your encouragement.

**Specific comments**

The manuscript should be considered for publication only after making minor revisions as follows:

Response: We thank the reviewer for the constructive comments, which are really important to improving our manuscript. We have carefully revised our manuscript as shown below.

1. Please clarify the implication and limitation of this study for air quality change. A three-dimensional circulation of typhoon consists of the rotational air flow in the horizontal direction and the in-up-out-down overturning flow in the vertical direction, along which air mass near the surface can rise into thunderstorm clouds, outflowing at high levels in the UTLS and subsiding in the periphery. To better discuss the link with three-dimensional circulation of typhoon, please reference this ACP paper: Jiang, Y. Zhao, T., Liu, J., et al. Why does surface ozone peak before a typhoon landing in southeast China? . Atmos. Chem. Phys., 2015, 15, 13331-13338.

Response: Thanks for the constructive comment and the important reference. We are deeply sorry that we ignored the three-dimensional structure of typhoons, especially the process of stratosphere-troposphere exchange of $O_3$. We have added this part, and reorganized the language in our revised manuscript. Please see lines 30-38 (Abstract), 410-422 (Sect. 3.3.2), 545-552 (Sect. 3.3.6), 634-641 (Sect. 4) and 728-730 (References) in the revised manuscript. Also, we have added a little implication to the discussion derived from the reference. Please see lines 564-565 (Sect. 3.3.6).

Figure R3 presents a fine map of temporal-vertical distribution of $O_3$ with vertical wind velocity over Shanghai, Hangzhou, Nanjing and Hefei between Typhoon Ampil and Typhoon Jongdari. Notably, $O_3$-poor air has penetrated into the stratosphere with the help of deep convection during typhoon, while $O_3$-rich air from the low stratosphere transported by downdrafts after typhoon. Combined with generated $O_3$ by photochemical reactions, high $O_3$ will appear in low troposphere. Moreover, the $O_3$ can remain at a high level in the residual at night, and can be transported to the surface by the second day. The transportation of $O_3$ in the upper layer plays an important role in the formation of $O_3$ pollution episodes on the surface.

[Figure]

[Figure]

[Figure]

[Figure]

Figure R3: Temporal-vertical distribution of O₃ with vertical wind velocity over Shanghai, Hangzhou, Nanjing and Hefei. The dotted purple line and solid blue line indicate the negative wind speeds (downward airflows) and positive wind speeds (upward airflows), respectively. The time refers to UTC. The black box indicates the area of stratosphere-troposphere exchange (STE).

2. All Figs. (a) and (b) in Figs 6, 8 and 9 are too tiny to identify. Please improve the Figure presentations with better quality.

Response: Thanks for the constructive comment. In our revised manuscript, we have enhanced the quality of these figures with higher resolution and larger font size. Please see lines 431-433 (Fig. 6), 475-477 (Fig. 8), 502-504 (Fig. 9) and 522-524 (Fig. 10) in the revised manuscript.

3. Lines 485-487: this statement that "The YRD region, which features a typical subtropical monsoon climate, is strongly influenced by the western Pacific subtropical high in summer." is incorrect or incomplete. The YRD, as a typical region of East Asian monsoon climate, is strongly influenced by typhoon activities over the Western Pacific.

Response: Thanks for the comment and correction. In this revising, the reviewer's suggestion is followed. "The YRD region, which features a typical monsoon climate, is strongly influenced by the western Pacific subtropical high in summer" has been changed to "The YRD, as a typical region of East Asian monsoon climate, is strongly influenced by typhoon activities over the Western Pacific.". Please see lines 530-534 (Sect. 3.3.6) in the revised manuscript.

4. In synoptics, the western Pacific subtropical high and tropical cyclone (Typhoon) over western Pacific are two large scale weather systems in opposition to each other. Therefore, please remove "The subtropical high", and "High temperature, southeast wind, downward airflow" from Fig. 11, also please correct the corresponding discussions.

Response: We appreciate this constructive comment. We are sorry that we did not carefully examine the synoptic meaning of the western Pacific subtropical high and tropical cyclone (Typhoon) in the last version, and we have changed both the schematic diagram and the corresponding discussions in the revised manuscript. Please see lines 44-45 (Abstract), 533-534 (Sect. 3.3.6), 557- 558 (Fig. 11) and 627 (Sect. 4) in the revised manuscript.

[revised manuscript text omitted]